# A Survey on Semantic Parsing

**Aishwarya Kamath**                    AISHWARYA.KAMATH@ORACLE.COM
*Oracle Labs*

**Rajarshi Das**                    RAJARSHI@CS.UMASS.EDU
*University of Massachusetts, Amherst*

## Abstract

A significant amount of information in today's world is stored in structured and semi-structured knowledge bases. Efficient and simple methods to query them are essential and must not be restricted to only those who have expertise in formal query languages. The field of semantic parsing deals with converting natural language utterances to logical forms that can be easily executed on a knowledge base. In this survey, we examine the various components of a semantic parsing system and discuss prominent work ranging from the initial rule based methods to the current neural approaches to program synthesis. We also discuss methods that operate using varying levels of supervision and highlight the key challenges involved in the learning of such systems.

## 1. Introduction

An important area of research cutting across natural language processing, information retrieval and human computer interaction is that of natural language understanding (NLU). Central to the objective of NLU is semantic parsing, which is framed as a mapping from natural language utterances to meaning representations. The meaning representation thus generated goes beyond shallow identification of roles and objects in a sentence to the point where it enables automated reasoning. It can be executed in a variety of environments in order to enable tasks such as understanding and executing commands for robotic navigation, data exploration and analysis by parsing natural language to database queries and query parsing in conversational agents such as Siri and Alexa.

Further, it is worthwhile to recognize that semantic parsing is inherently different from other sequential prediction tasks such as machine translation and natural language generation in that unlike the latter, it involves prediction of inherently structured objects that are more tree-like. Additionally, it has to also adhere to certain constraints in order to actually execute in a given environment, introducing unique challenges.

The contributions of this survey are firstly, to introduce the key components of the semantic parsing framework. Secondly, to make an uninformed reader comfortable with the field by providing an intuition for how it has developed through the years. Thirdly, to contrast various methodologies that have been followed in terms of the level of supervision, types of training strategies, amount of feature engineering and ways to incorporate structural constraints.

To make a reader familiar with the terminology used, (§ 2) formally introduces the task of semantic parsing along with its fundamental components. In (§ 3) we will explore the

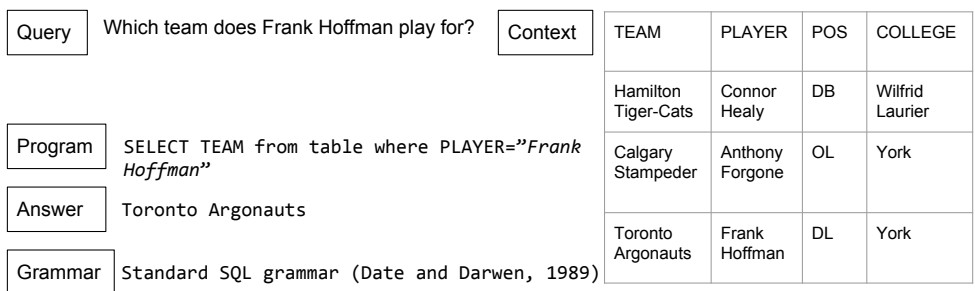

Figure 1: Example of a semantic parsing task with various components.

field from its early days where systems were based on hand crafted rules, restricting them to narrow domains and trace its path through times where we learned statistical models on examples of input-output mappings. The mapping to logical forms typically relies heavily on the availability of annotated logical forms and difficulties in procuring such labeled data gave rise to attempts at using alternative methods of training involving weak supervision. (§ 4) describes the transition to using data sets that consist of questions annotated only with the final denotation, giving way to new styles of training. (§ 5) details alternative forms of supervision. (§ 6) describes the adoption of encoder decoder frameworks to model the semantic parsing task. The three common learning paradigms used to learn the parameters of the model are explained in (§ 7).

## 2. Task and Salient Components

Semantic parsers map *natural language* (NL) utterances into a *semantic representation*. These representations are often (interchangeably) referred to as logical forms, meaning representations (MR) or programs (§ 2.1). The representations are typically executed against *an environment* (or context) (§ 2.3) (e.g. a relational knowledge base) to yield a desired output (e.g. answer to a question). The meaning representations are often based on an underlying formalism or *grammar* (§ 2.2). The grammar is used to *derive* valid logical forms. A *model* is used to produce a distribution over these valid logical forms and a *parser* searches for high scoring logical forms under this model. Finally, a *learning algorithm* (§ 7) is applied to update the parameters of the model, by using the training examples. As a concrete example, consider the following from the WikiSQL dataset [Zhong et al., 2017]. The goal of the semantic parser in figure 1 is to generate a valid SQL program (that can be derived from the grammar [Date and Darwen, 1989]) given the natural language query and which produces the correct answer when executed against the table (context). Most published works also assume access to a deterministic executor (e.g. SQL interpreter) that can be called as many times as required.

### 2.1 Language for Meaning Representation

The first main component of a semantic parsing framework is the language of the logical form or meaning representation.

**Logic based formalisms**: A straightforward language representation is one that uses *first order logic* (FOL) — consisting of quantified variables and functions that can only take objects as arguments. Following the examples from Liang [2016], simple statements

such as "ALL PRIMES GREATER THAN 2 ARE ODD" can easily be expressed in first order logic $\forall x.\texttt{prime(x)}\land\texttt{more(x,2)}\rightarrow\texttt{odd(x)}$. However FOL has limitations in its inability to represent and manipulate sets, for e.g. "COUNT THE NUMBER OF PRIMES LESS THAN 10" cannot be expressed just with FOL. However, FOL augmented with *lambda calculus* (LC) [Carpenter, 1997] can be used to represent this — $\texttt{count}(\lambda x.\texttt{prime(x)}\land\texttt{less(x,10)})$, Here, $\lambda x$ denotes the *set* of all x that satisfies a given condition. More details on the constituent units of lambda calculus can be found in [Artzi et al., 2013]. LC has been used for querying databases [Zettlemoyer and Collins, 2005], in implementing conversational agents [Artzi and Zettlemoyer, 2011] as well as for providing instructions to robots [Artzi and Zettlemoyer, 2013]. *Lambda Dependency based Compositional Semantics* ($\lambda$-DCS) by Liang [2013], is a more compact representation of LC. In this formalism, existential quantification is made implicit, allowing the elimination of variables. An example of this is "PEOPLE ABOVE THE AGE OF 20" would be $\lambda x.\exists y.\ \texttt{AgeGreaterThan(x,y)}\land\texttt{Age(y, 20)}$ in LC and $\texttt{AgeGreaterThan.Age.20}$ in $\lambda$-DCS. Readers are directed to [Liang, 2013] for details. Several works such as [Berant et al., 2013, Pasupat and Liang, 2015, Krishnamurthy et al., 2017] use this formalism.

**Graph based formalisms**: Here the semantic form of a NL utterance is represented as a labeled graph where the nodes usually denote entities/events and edges denote semantic relations between the nodes. Graph representation offers a few advantages over other representations — (a) They are easy for humans to read and comprehend, when compared to e.g. a `SQL` program. (b) They tend to abstract away from the syntactic structure and can even not be anchored to the words in the sentence. For example in Abstract Meaning Representation (AMR) [Banarescu et al., 2013], the word "graduation" can be mapped to the sense "`graduate-01`" in Propbank [Kingsbury and Palmer, 2002]. (c) There exists a rich literature on graph algorithms which can be leveraged for learning. Examples of graph based formalisms are AMR, Universal Conceptual Cognitive Annotation (UCCA) [Abend and Rappoport, 2013], Semantic Dependency Parsing [Oepen et al., 2015] etc. Recently, Kollar et al. [2018] introduced the Alexa Meaning Representation language that aims to capture the meaning of spoken language utterances.

**Programming languages**: Of late, there have been efforts to convert a natural language query directly to high-level, general purpose programming languages (PL) such as Python, Java, SQL, Bash. Converting into general purpose PLs has its advantages over converting into structured logical forms since (a) PLs are widely used in the developer community, (b) the structured logical forms have relatively simple schema and syntax, limiting their scope and are generally developed for a particular problem, making them domain specific. In terms of application, being able to generate PLs directly from natural language queries would result in smarter programming environments for developers, and more importantly would allow non-experts to integrate non-trivial software components in their products. This motivation is also in spirit with the emerging research area of automated machine learning (AutoML) [Feurer et al., 2015].

## 2.2 Grammar

The grammar is a set of rules whose function is to define a set of candidate derivations for each natural language utterance. The type of grammar used decides the expressivity of the

semantic parser as well as the computational complexity associated with building it. When dealing with complex but well structured queries (e.g. GeoQuery [Zelle and Mooney, 1996]), strict grammars like *Combinatory Categorial Grammar*[1] [Steedman, 1996] having many rules provide computational advantage by virtue of generating fewer possible derivations [Zettlemoyer and Collins, 2005, 2007, Artzi and Zettlemoyer, 2011, Kwiatkowski et al., 2013]. On the other hand, when the language is noisy, *floating rules* have the advantage of being able to capture ill formed sentences such as those produced when dealing with crowd sourced utterances [Wang et al., 2015], at the cost of producing many more candidate derivations [Berant and Liang, 2014, Pasupat and Liang, 2015]. Recent work on general purpose code generation (such as SQL, Python etc) leverage the well-defined grammar associated with the programming language by attempting to directly generate the Abstract Syntax Trees associated with them (§ 6.3).

## 2.3 Underlying context

All mappings from natural language to semantic representations are with respect to the underlying context or environment on which they are based. An example of this context is a knowledge base (e.g. Freebase [Bollacker et al., 2008]) which is essentially a store of complex structured and unstructured data. The context can also be a small knowledge store — tables from Wikipedia [Pasupat and Liang, 2015], spreadsheets [Gulwani et al., 2012], images [Suhr et al., 2017] etc. — or designed for special purpose application such as flight-booking [Hemphill et al., 1990], geography related queries [Zelle and Mooney, 1996] and robot navigation [MacMahon et al., 2006, Artzi and Zettlemoyer, 2013].

## 3. Early attempts

### 3.1 Rule based systems

Early semantic parsing systems were mainly rule-based, making them domain specific. These can be based on pattern matching such as SAVVY [Johnson, 1984] which although simple, is quite brittle due to the shallow nature of the pattern matching. Another methodology employs syntax-based systems. An example of this kind of system is LUNAR [Woods, 1973] where a syntactic parser generates a parse tree that is mapped using rules to an underlying database query language. A stricter grammar that takes into account the semantic categories instead of just syntactic are used in semantic-grammar systems such as [Thompson et al., 1969, Waltz, 1978, Hendrix et al., 1978, Templeton and Burger, 1983]. Building the parse tree in this type of setting involves having non terminal nodes as semantic categories which constrain the possible assignments to the tree in such a way that the mapping from the tree to the query language is easier. Although it streamlines the flow for a given task, this inclusion of semantic knowledge of a specific domain makes it even harder to adapt to other domains. This is in contrast to the syntax-based grammars where it is possible to at least re-use generic syntactic information across domains. An in depth compilation of the various NLIDB systems prior to the 90s can be found in [Androutsopoulos et al., 1995].

---

1. An accessible tutorial on parsing with CCGs can be found at https://yoavartzi.com/tutorial/

### 3.2 Rise of statistical techniques

The seminal work of Zelle and Mooney [1996] introduced a model which is able to train on a corpus of pairs of sentences and database queries. The system employed a deterministic shift reduce parser and developed an algorithm called CHILL which is based on Inductive Logic Programming. The previous work that had used corpus based training were all focused on shallow parsing, using corpora that had to be previously annotated with syntactic information. The evaluation of these systems was also imperfect as they were compared based on simulated metrics of parsing accuracy. On the other hand, Zelle and Mooney [1996] build a framework which converts the natural language sentence to a database query, facilitating straightforward evaluation: by implementing the query on the database, one can check if the answer returned is satisfactory. The operators used in this framework require a semantic lexicon as a-priori knowledge. Thompson and Mooney [2003] improve on this by learning a lexicon of phrases paired with meaning representations, from the data. Experimental results showed that a parser using the learned lexicon performs comparably to one that uses a hand built lexicon. This data driven approach also allowed lexicons and parsers to be easily built for multiple languages — Spanish, Japanese and Turkish. However, even Zelle and Mooney [1996] required annotated examples where the annotation effort is not trivial. To help with this, Thompson et al. [1999] suggest looking at alternative training procedures which involve using active learning in order to reduce the number of annotated examples required to solve complex tasks.

A consistent theme through these times was the increasing use of statistical learning techniques that are able to learn models on given examples of desired input output pairs. Among the fully supervised attempts at semantic parsing using fully annotated sentence-logical form pairs are [Zelle and Mooney, 1996, Zettlemoyer and Collins, 2005, 2007, Kwiatkowski et al., 2010]. In Zettlemoyer and Collins [2005], the learning algorithm takes as input pairs of sentences and their respective lambda-calculus expressions. An example of this kind of pair from the GeoQuery domain is : What states border Texas :$\lambda x.state(x)borders(x, texas)$. Given a CCG lexicon L, there would be many different parses that could result for a given sentence. To deal with this ambiguity, a PCCG (Probabilistic CCG) defines a method for ranking possible parses for a given sentence in terms of the probability. More concretely, a log linear model that is characterized by a feature vector $f$ and a parameter vector $\theta$, is usually used to select the most likely parse. For a given sentence $s$, parse $z$ and logical expression $l$, the log linear model is defined as :

$$P(z, l|s; \theta, L) = \frac{e^{\theta f(s,z,l)}}{\sum_{(z',l')} e^{\theta f(s,z',l')}}$$

The features could be as simple as counts of certain lexical entries in the current parse. The parsing (inference) part of the process involves finding the most likely logical form $l$, given an input sentence $s$ when the parameters and the lexicon are known:

$$F(x) = \arg \max_l p(l|s; \theta, L) = \arg \max_l \sum_z P(z, l|s; \theta, L)$$

obtained by marginalizing over all the latent parses $z$. If the feature vectors only operate locally, it is possible to employ dynamic programming approaches such as those employed

in CKY parsing algorithms used for finding the most likely parse in probabilistic context free grammars [Manning et al., 1999]. Beam search is used during parsing to remove low probability parses, increasing efficiency.

In Zettlemoyer and Collins [2005], the lexicon is induced as part of the learning procedure by only preserving lexical entries that result in the required parse. However such a system would have issues due to the inflexibility of CCG when it is applied to raw natural language inputs, having variable ordering of words, colloquial language and sometimes even missing words. Zettlemoyer and Collins [2007] tweak the learning algorithm so that it is possible to deal with these kinds of variations. This is carried out by using extra combinators that are not standard CCG, by relaxing some parts of the grammar such as the ordering of words along with having learned costs for these new operations. They also describe a novel online algorithm for CCG learning which employs perceptron training of a model with hidden variables, while also inducing grammar in an online fashion. Meanwhile, Kwiatkowski et al. [2010] aim to have a more general framework that is agnostic to the choice of natural language as well as logical expressions, using a CCG, followed by higher order unification which allows the model to search the space of all grammars consistent with training data, while also allowing estimation of the parameters for the log-linear parsing model.

However, Zettlemoyer and Collins [2005, 2007] had deficiencies in the form of data sparsity caused by not sharing weights for related classes of words. Kwiatkowski et al. [2011] improve upon this by introducing factored lexicons. The main intuition behind this work is that within a particular class of words, the variation between the lexical items is *systematic*. Decomposing the lexical entries allows sharing of information between multiple lexical entries, easing the issue of data sparsity. They address this by splitting the lexicon into **lexemes** which are mappings from natural language words or phrases to a set of logical constants and **lexical templates** which are pre-defined templates used to capture the variable usage of words. This decomposition results in a more concise lexicon which also requires less data to learn from. Subsequent work using CCGs mostly employ this factored lexicon approach.

A consequence of the fact that these systems require complex annotation is that they are hard to scale and they only work in narrow domains. Generating the data sets using manual annotation is still a formidable task and due to this, supervision in various different forms has been explored. The next two sections list several of these attempts.

## 4. Semantic parsing from denotations

The next transition came from training systems solely on the **result** of the execution (a.k.a. denotation) of the formal meaning representation (program), in the absence of access to the program itself. This can take many forms, e.g. the result of executing a query on a KB, the final state of a robot that is given instructions, etc. Training models with such **weak supervision** poses certain challenges. The first is an extremely **large space** of potential programs that have to be explored during training. The second is the noise that is inevitable in these systems due to **spurious programs** that lead to the correct answer by chance.

Berant et al. [2013] learn a semantic parser from question answer pairs on Freebase. A major problem while using such large KBs is that there exist a very large number of logical predicates for a given question. They use a two step process to deal with this issue: they

first have a coarse grained solution by aligning the sentences to predicates of the KB by asserting that a phrase and a predicate align if they co-occur with many of the same entities. They then have a bridging operation which provides additional predicates by looking at the neighbors. This is especially helpful when the predicates are expressed weakly or implicitly. Concurrent work by Kwiatkowski et al. [2013] also employs a two stage approach which involves first using a CCG to map utterances to an abstract under-specified form which is domain independent. This abstraction allows sharing across different domains (§ 6.1), instead of having to learn the rules for each target domain. This is followed by an ontology matching step wherein the domain specific knowledge is incorporated using information about the target ontology as well as lexical similarity between constants that are part of the under specified form and the domain specific constants. A linear model over all derivations is learned, involving the CCG parsing steps as well as the ontology matching, with supervision only in the form of final answers to the questions.

Pasupat and Liang [2015] use question-answer pairs as supervision to tackle the task of answering complex questions on semi-structured tables. They use a realistic setting where the train and test tables are disjoint so that the model should be able to generalize to entities and relations that were not seen at training. A consequence of this is that the regular approach of building a lexicon with mappings from phrases to relations cannot be built and stored. Further, due to the complex nature of the questions that involve operations such as aggregations and comparisons, the search space of all possible logical forms grows exponentially. To combat the issue with the unseen tables, their approach first builds a knowledge graph from the table in question resulting in an encoding of all the relevant relations. Using the information from this graph, candidate parses of the question are generated and these candidates are ranked using a log linear model, whose parameters are learned using MML (§ 7.1). To deal with the exploding search space, they use beam search with strong type and denotation constraints along with strict pruning of invalid or redundant logical forms.

In the context of semantic parsing for robot navigation, the training data consists of triples of a natural language instruction, start state and validation function that specifies whether the set of actions produced the expected result. Artzi and Zettlemoyer [2013] use a grounded CCG by incorporating the execution of the logical form in each state to drive the CCG parsing. This joint inference procedure allows the parser to leverage cues from the environment, making it possible to learn while executing the instructions.

## 5. Alternate forms of supervision

**Auxiliary syntactic information**: Krishnamurthy and Mitchell [2012] are able to train a semantic parser without needing any sentence-level annotations. Instead, they combine readily available syntactic and semantic knowledge to construct a versatile semantic parser that can be applied to *any* knowledge base. Using just the knowledge base and automatically dependency parsed sentences from a web corpus, they are able to generate the correct logical form with 56% recall. Using dependency parse based heuristics, a lexicon is generated for relation instances that occur in the web based corpus. A mention identification procedure links these to entities in the knowledge base resulting in triples consisting of sentences, along with two identified mentions. By matching the dependency path between the two entity

mentions to a set of pre-identified patterns, lexical entries are generated. This lexicon is used by the CCG parser with allowances to skip words that do not occur in the vocabulary of the knowledge base — ensuring that the logical forms contain only predicates from the knowledge base. The resulting parser is able to handle complex constructs having conjunctions of various categories as well as relations that share arguments.

**Unsupervised**: [Poon and Domingos, 2009] was the first attempt at learning a semantic parsing model in an unsupervised manner. They begin by clustering tokens that are of the same type, followed by recursively combining sub-expressions that belong to the same cluster. Their model learns to successfully map active and passive voices to the same logical form showing the robustness of their approach. They evaluate their model on a fact prediction task on a KB that they extract from the GENIA [Kim et al., 2003] biomedical corpus. A major problem with the approach of generating self induced clusters for the target logical forms is the mismatch with existent ontology and databases. As a result, their approach is unable to answer complex questions on existing databases without an extra ontology matching step. Instead, Poon [2013] exploit the database as a form of indirect supervision, allowing them to attack more complex questions. Here, they combine the unsupervised semantic parsing approach with grounded learning using the database. A probabilistic grammar is learned using EM and the main merit of their method is that the search space is constrained using the database schema making learning much easier and also ensures that the parses that are generated can be readily executed on the database.

**Supervision from conversations**: An unusual source of supervision for semantic parsing systems is from conversations. Artzi and Zettlemoyer [2011] use conversational feedback in order to learn the meaning representation for user utterances. They induce semantic parsers from un-annotated conversational logs. They posit that it is often possible to decipher the meaning of an utterance by analysing the dialog that follows it. They use a PCCG for inducing the grammar and by using a loss sensitive perceptron algorithm, the model obtains a rough estimate of how well a possible meaning for an utterance matches with what conversation followed after it as well as how much it matches our expectations of what is generally said in the dialog's domain.

**Visually grounded semantic parsing**: Recent work by Ross et al. [2018] involves learning a semantic parser using captioned videos. Using a factored lexicon approach as in [Kwiatkowski et al., 2011], they employ a weighted linear CCG based semantic parser that uses supervision in the form of a visual validation function. This function provides a sense of the compatibility of a parse with a given video and is the only supervision used to drive the lexicon generation process in the modified GENLEX [Artzi et al., 2013]. While the parser is learned using the videos, they are not used at test time.

**Using canonical forms**: Wang et al. [2015] demonstrate how to train a semantic parser in a domain starting with zero training examples. Their framework has two parts - the first is a *builder* that provides a seed lexicon containing a canonical form for each predicate, given the target database schema. The second part is a *domain general grammar* that uses predicate-canonical form pairs from the seed lexicon to generate pairs of logical expressions and *canonical utterances*, which are ill-formed natural language-like sentences preserving the semantics of the corresponding logical expressions. Crowd workers are then employed to paraphrase these canonical utterances to actual natural language utterances to build the data set. This method has the advantage of being *complete* in the sense that the entire

logical functionality of the grammar is reflected in the data set due to the reverse way in which it is built. This is in contrast to other approaches where the questions and answers in the data set dictate the abilities of a semantic parser. For example, the WebQuestions dataset does not have any instances of questions with numerical answers, resulting in any semantic parser trained on this data set to also not be able to handle such cases. It is also an easier effort to generate annotations for paraphrases than actual answers to the questions, reducing the burden on the crowd worker and making the cost of the jobs lesser.

**Machine translation techniques**: Wong and Mooney [2006] employ statistical machine translation (MT) techniques for the task of semantic parsing by arguing that a parsing model can also be viewed as a syntax-based translation model. They achieve good performance and develop a model that is more robust to word order. The authors tried to depart from the more deterministic parsing strategies that were prevalent at the time [Zelle and Mooney, 1996, Kate et al., 2005] and only assume access to an unambiguous, context free grammar of the target logical form. Building on the work by [Kate et al., 2005], Wong and Mooney [2006] use a statistical word alignment algorithm as seen in [Brown et al., 1993] in order to get a bilingual lexicon which has pairings of the natural language sentence with its respective translation in the logical form. The full logical form is obtained by combining all the input sub strings and their translations. Andreas et al. [2013] explicitly model semantic parsing as a MT task and achieve results comparable to the state of the art on GeoQuery at the time. In order to use MT techniques, they linearize the meaning representation language (MRL) by performing a pre-order traversal of each function and label every token with the number of arguments it takes, which is later used during decoding to distinguish well-formed MRs. It also helps to model functions that can take multiple number of arguments during alignment, depending on the context. Both phrase based and hierarchical translation models are used to extract pairs of source↔target aligned sentences. A language model is learned on the MR training data so that the model learns which arguments are more favoured by particular predicates of the MRL. The success of this work demonstrated that MT methods should be considered as competitive baselines.

**Learning from user feedback**: Iyer et al. [2017] present an approach to improve a neural semantic parser based on user feedback wherein a rudimentary parser is used to bootstrap, followed by incorporating binary user feedback to improve the parse. In Lawrence and Riezler [2018], the semantic parser is improved using interaction logs, available in abundance as a result of commercially deployed systems logging a huge amount of user interaction data, and employs a form of counterfactual learning [Bottou et al., 2013]. They show improvements on the NLMAPS dataset in the OPENSTREETMAP domain. Although this work is specifically designed to work on a small system it a promising avenue especially for industry due to the extensive availability of logged data.

## 6. Enter Seq2Seq

Over the last couple of years, there has been a rise in end to end approaches for semantic parsing using encoder-decoder frameworks based on recurrent neural networks. This approach has been largely successful in a variety of tasks such as machine translation [Kalchbrenner and Blunsom, 2013, Cho et al., 2014, Sutskever et al., 2014], syntactic parsing [Vinyals et al., 2015b], image captioning [Vinyals et al., 2015c], etc. In the case of semantic

parsing, it consists of learning a mapping from the natural language utterance to the meaning representation, avoiding the need for intermediate representations. This flexibility has advantages as well as disadvantages- it alleviates the need for defining lexicons, templates and manually generated features, making it possible for the models to generalize across domains and meaning representation languages. However, traditional approaches are able to better model and leverage the in-built knowledge of logic compositionality.

Dong and Lapata [2016] set this up as a sequence transduction task where the encoder and decoder are in the same format as defined by Sutskever et al. [2014], employing L-layer recurrent neural networks (LSTMs in this work) that process each token in the sequence. In order to better model the **hierarchical structure** of logical forms and make it easier for the decoder to keep track of additional information such as parentheses, the authors propose to use a decoder that takes into account the tree structure of the logical expression. This is enforced by having an action (a special token) for generating matching parentheses and recursively generating the sub-tree within those parentheses. The decoder is provided with additional **parent feeding connections** which provides information about the start of the sub-tree, along with soft attention to attend over tokens in the natural language utterance as in Bahdanau et al. [2014] and is able to perform comparably against previous explicit feature based systems.

A fundamental concern regarding these methods is their inability to learn good enough parameters for **words that are rare** in the dataset. The long tail of entities in these small datasets raises the need for special pre-processing steps. A common method is to **anonymize the entities** with their respective types, given an ontology. For example towns such as Amherst are anonymized as `Town`, and later post processed to fill in the actual entity. Another approach to counter this is to use Pointer Networks [Vinyals et al., 2015a] where a word from the input utterance is directly copied over to the output at each decoding step. A combination of these two referred to as **attention-based copying**, allowing for more flexibility so that the decoder can either choose to copy over a word or to pick from a softmax over the entire vocabulary is employed in [Jia and Liang, 2016]. This work also introduces an approach which can be classified as a data augmentation strategy. They propose to build a generative model over the training pairs (natural language to logical expression) and sample from this to generate many more training examples, including those of increased complexity. The model used to do the actual parsing task is the same as in [Bahdanau et al., 2014] and the main contribution of this work is the emphasis on a conditional independence property (which holds most of the time) that is fundamental to compositional semantics- *the meaning of any semantically coherent phrase is conditionally independent of the rest of the sentence.* This allows them to build a synchronous context free grammar (SCFG) which the generative model samples from, to provide a distribution over pairs of utterances and logical expressions. We refer the reader to [Jia and Liang, 2016] for more details on the grammar induction steps. This approach boosts performance, due in part to the more complex and longer sentences that are generated as part of the data recombination procedure, forcing the model to generalize better.

## 6.1 Employing Intermediate Representations

In closed domains, learning mappings from lexical items to all program constants becomes cumbersome and introducing a degree of indirection has been shown to be beneficial. Disen-

tangling higher and lower level semantic information allows modelling of meaning at varying levels of granularity. This is profitable as the abstract representation is more likely to generalize in examples across small data sets. Further, using this abstraction helps to reduce the search space of potential programs as well as the number of false positives that are generated due to spurious programs that can generate the same denotation (answers or binary labels) just by chance.

Although frameworks based on the encoder decoder framework provide greater flexibility, they generally do not allow interpretation and insights into meaning composition. A two stage approach such as that employed by Cheng et al. [2017] aims to build intermediate structures that provide insight into what the model has learned. Further they use a transition based approach for parsing which allows them to have non-local features - an advantage over other methods using chart parsing which require features to be decomposed over structures in order to be feasible.

Goldman et al. [2017] utilize a fixed KB schema to generate a smaller lexicon that maps the more common lexical items to typed program constants. Here, due to the abstract representation being confined to the domain of spatial reasoning, they are able to specify seven abstract clusters of semantic types. Each of these clusters have their own lexicon having language to program pairs associated with the cluster. A rule based semantic parser then maps the abstract representation created to one of the manually annotated examples, if they match. To counter the limited applicability as a result of the rule based parser, the authors propose a method to augment their data by automatically generating programs from utterances by using manually annotated abstract representations. In this way, they can choose any abstract example and then instantiate the clusters that are present in it by sampling the utterance to program pairs which are in that abstract cluster. These generated examples are used to train a fully supervised seq2seq semantic parser that is then used to initialize the weakly supervised model.

A strong merit of these lifted intermediate representations is that they allow **sharing between examples**, leading to easier training. Similar methods to generate intermediate templates have been employed in [Dong and Lapata, 2018] where a *sketch* of the final output is used for easier decoding. These methods have the added advantage of having an idea of what the intended meaning for the utterance is at a high level, and can exploit this **global context** while producing the finer low level details of the parse.

### 6.2 Decoding using a constrained grammar

By explicitly providing the decoder with constraints in the form of a target language syntax, Yin and Neubig [2017] alleviate the need for the model to discover the intrinsic grammar from scarce training data. The model's efforts can then be better concentrated on building the parse, guided by the known grammar rules. The natural language utterance is transduced into an Abstract Syntax Tree (AST) for a given programming language which is then converted to the programming language deterministically. Generation of the AST takes place through sequential application of *actions*, where the set of possible actions at each time step incorporates all the information about the underlying syntax of the language. Xiao et al. [2016] similarly employ the grammar model as prior information so that the generation of derivation trees is constrained by the grammar. By explicitly ensuring that the

decoder's prediction satisfy *type constraints* as provided by a type-constrained grammar, Krishnamurthy et al. [2017] are able to perform significantly better than the Seq2Tree model from [Dong and Lapata, 2016] showing that it is not enough to just generate well-formed logical expressions, but having them satisfy type constraints is also important. They also employ entity linking to better map the question tokens to entities that they refer to. By using features such as NER tags, edit distance and lemma matches, they are able to better handle the problem of rare entities, for which it is difficult to learn good embeddings.

### 6.3 General Purpose Code Generation

**Data.** Training such models requires a parallel corpus of natural language utterances and their corresponding PL form. Although such data sources are not readily available, one can employ distant supervision techniques to existing open source code bases (e.g. `github`[2]) and forums (e.g. `StackOverflow`[3]) to gather such aligned data. For example, natural language descriptions can be mined from code comments, documentation strings (docstrings), commit messages, etc and can be aligned to the code. Ling et al. [2016] introduced two datasets by aligning the open-source Java implementation of two trading card games – HEARTHSTONE and MTG to the description in the playing cards. Similarly, Barone and Sennrich [2017] develop a large dataset of Python functions aligned to their docstrings and Iyer et al. [2018] introduce the CONCODE dataset where both method docstrings and an initial snippet of code context is aligned to the implementation of a method in a Java class. Other datasets are introduced in [Iyer et al., 2016, Yao et al., 2018, Lin et al., 2018, Oda et al., 2015].

**Methodology.** Most contemporary works design the problem of code generation in the encoder-decoder (seq2seq) framework. The various datasets have their idiosyncracies, e.g. SQL and Bash datasets are generally single line of source code and hence the target sequence is short in length. On the contrary in HEARTHSTONE and MTG, models have to generate an entire class structure given the natural language description.

The encoder module is usually a variant of bidirectional recurrent neural network architecture where the input consists of word level tokens [Yin and Neubig, 2017] or characters [Ling et al., 2016]. The encoder module of Rabinovich et al. [2017] takes advantage of the structure present in the input (e.g. various textual fields of each cards in the HEARTHSTONE dataset) and encodes each component separately.

Since PLs have a well-defined underlying syntax, the decoder module presents more interesting challenges. Initial work by Ling et al. [2016] ignores any such syntactic constraints and lets the model learn the syntax from data. As a result, there is a possibility they may produce syntactially ill-formed code. Later work by Yin and Neubig [2017], Rabinovich et al. [2017] and Iyer et al. [2019] explicitly model the underlying syntax of the PL in the decoding stage and hence can guarantee well-formed code. Similarly, Sun et al. [2018] develop a model which explicitly encodes the table structure and SQL syntax for the task of sequence to SQL generation. In Rabinovich et al. [2017], the decoder module decomposes into several classes of modules and these are composed to generate the *abstract syntax trees* (ASTs) in a top-down manner. Learning corresponds to learning the sequence of decisions

---

2. http://github.com

3. http://stackoverflow.com

and is guided by the ASTs during training. This work can be seen as complimentary to neural module networks [Andreas et al., 2016b,a, Hu et al., 2017].

## 7. Learning

### 7.1 Maximum marginal likelihood (MML)

In the **fully supervised setting** where the training set consists of $\{(x_i, y_i, c_i) : i = 1 \ldots n\}$, with access to the natural language utterance $x_i$, the *fully annotated output logical form $y_i$* and the context $c_i$, this objective is of the form:

$$J_{MML} = \log p(y_i | x_i, c) = \log \sum_{d \in D} p(y_i, d | x_i, c)$$

where D is the set of all possible derivations that produce the logical form $y_i$. These are generated as part of the parsing procedure and are not available as part of the training set. These derivations are treated as latent variables and marginalized out of the objective.

In the **weakly supervised setting**, when our training data is of the form $\{(x_i, z_i, c_i) : i = 1 \ldots n\}$, where $z_i$ is the denotation obtained after executing the latent logical form $y_i$ on the context $c_i$, the objective is of the form:

$$J_{MML} = \log p(z_i | x_i, c) = \log \sum_{y \in Y} p(z_i | y, c_i) p(y | x_i, c_i)$$

In this case, we only need to check whether the denotation obtained as a result of the execution of a given $y \in Y$ (where Y is the set of all programs) on $c$ is equal to $z$. Depending on this, the first term of the summation is either a 1 or 0. Further, considering $Y^*$ as the set containing only the programs that execute to the correct denotation $z$, we obtain the following:

$$J_{MML} = \log \sum_{y \in Y^*} p(y | x_i, c_i)$$

In practice, the set of y that are considered is usually approximated using beam search. The search space is typically pruned using strong type constraints. The search procedure can be done as part of the learning in an *online* manner, or executed beforehand in an *offline* manner, which we describe next.

**Online search** — Several works in semantic parsing literature employ MML with online search [Pasupat and Liang, 2015, Liang et al., 2011, Berant et al., 2013, Goldman et al., 2017]. Including the search process as part of the learning necessitates running a beam search with a wide beam (for example $k$=500 in [Berant et al., 2013]), leading to expensive back propagation updates. The set of candidate logical forms keep changing in this approach and there is still no guarantee that every logical form that executes to the correct denotation will be included in the beam.

**Offline search** — In this setting, a set of logical forms that result in the correct final denotation are automatically generated beforehand for each example. Krishnamurthy et al. [2017] use this procedure referred to as *dynamic program on denotations* (DPD)[Pasupat and Liang, 2016], which leverages the fact that the space of possible denotations grows much more slowly than that of logical forms, since many logical forms can have the same

denotation. This enumeration makes it possible to have smaller beam sizes as a result of collapsing together many logical forms which have the same denotation. Further, DPD is guaranteed to recover all consistent logical forms up until some bounded size.

## 7.2 Reinforcement learning

In this learning paradigm, an agent is considered to make a sequence of decisions based on a policy and receives a reward $R(y, z)$ for a given program $y$ and answer $z$ at the end of the sequence if the sequence executes to the correct denotation z, and 0 otherwise. Typically, a stochastic policy is trained with the goal of maximizing the expected reward. The expected reward for an example $(x_i, z_i, c_i)$ can be written as:

$$J_{RL} = \sum_{y \in Y} p(y|x_i, c_i) R(y, z_i)$$

Since the set of all possible programs is large, the gradient is computed by sampling from the policy distribution. This is problematic when the search space is large and special techniques are employed to improve sampling. [Liang et al., 2016, Zhong et al., 2017, Andreas et al., 2016a, Misra et al., 2018, Das et al., 2017].

An issue with both of the above approaches is that they are based on exploration as a function of the model's current policy. This means that if a correct program is incorrectly assigned a low probability, it will fall out of the beam, or in the RL version, not be further sampled, further down-weighting the probability of this program. The same applies for falsely high probability *spurious* programs that incorrectly gain even more importance. Further, the REINFORCE [Williams, 1992] method has the added disadvantage that in the case of peaky distributions, i.e., a single program accounts almost entirely for the probability under the current policy, it will repeatedly sample the mode of the distribution. On the other hand, MML based exploration using beam search will always use the N-1 spaces in its beam to explore programs other than the highest scoring one.

[Guu et al., 2017] demonstrate how to overcome this issue by suggesting an $\epsilon$-greedy randomized beam search that gets the best of both worlds. In this approach, they follow regular beam search to compute all possible continuations and rank them in descending order according to their probabilities. But, instead of just picking the top $k$ highest scoring continuations, they are uniformly sampled without replacement with probability $\epsilon$ and retrieved from the remaining pool according to their rank, with probability $1 - \epsilon$. In a more principled approach to solving the exploration problem while also ensuring good sample efficiency, [Liang et al., 2018] propose a method that employs an external memory buffer that is used to keep track of high reward trajectories. They propose to also pick trajectories by rejection sampling using the memory buffer in addition to sampling from the buffer. This ensures that while high reward trajectories are not forgotten, it avoids giving too much importance to possibly spurious high reward trajectories, hence reducing the bias. Further, by keeping track of the high reward trajectories using the memory buffer, they are able to achieve a reduction in variance of the estimates by a factor that depends on the size of the buffer. They are able to outperform the previous state of the art on WikiTableQuestions.

### 7.3 Maximum margin reward methods (MMR)

In this third learning model, training usually involves maximizing a margin or minimizing a risk. Similar to the latent variable structured perceptron [Zettlemoyer and Collins, 2007] and its loss sensitive variants, this training paradigm involves structured output learning. Given a training set $\{(x_i, z_i, c_i) : i = 1...n\}$, this method first finds the highest scoring program $y_i^*$ that executes to produce $z_i$ from the set of programs returned by the search. Considering $y^*$ is the highest scoring program, we can find the set of all violating programs as:

$$V = \{y'|y' \in Y \text{ and } score_\theta(y^*, x, c) \le score_\theta(y', x, c) + \delta(y^*, y', z)\}$$

where the margin $\delta(y^*, y', z) = R(y^*, z) - R(y', z)$. From this, we obtain the most violating program as

$$\overline{y} = \underset{y' \in Y}{\operatorname{argmax}}\{score_\theta(y', x, c) - score_\theta(y^*, x, c) + \delta(y^*, y', z)\}$$

So for a given example $(x_i, z_i, t_i)$ the max margin objective or negative margin loss as:

$$J_{MMR} = -max\{0, score_\theta(\overline{y_i}, x_i, t_i) - score_\theta(y_i^*, x_i, c_i) + \delta(y_i^*, \overline{y_i}, z_i)\}$$

Where only the score of the highest scoring program and the one that violates the margin the most, are updated. Iyyer et al. [2017] and Peng et al. [2017] use this type of framework. Misra et al. [2018] explain how these various training paradigms relate to each other and provide a generalized update equation making it easier to choose an approach that would best suit a particular problem.

## 8. Future directions

Despite significant progress over the years, there still exist multiple directions worth pursuing. Integrating confidence estimation in the responses from a semantic parser has been largely unexplored and we believe is a useful direction to pursue. In order to have meaningful interactions with conversational agents, knowing when the model is uncertain in its predictions allows for adding human in the loop for query refinement and better user experience. Another promising direction is building semantic parsers that perform well across multiple domains, especially useful when the model needs to be deployed to various user domains that are disparate and lacking sufficient labelled data. Multi-task learning, co-training and fine-tuning across datasets in order to leverage shared representational power as well as being able to generalize to new domains could be a direction to pursue. Leveraging known structure in smarter ways in order to reduce the representational burden on the model and also help it generalize better is currently an area of great interest in the community, with scope for innovation. Since the logical form representation of an utterance is a semantic interpretation of the query, studies for natural language understanding such as RTE could look into leveraging these forms as part of their models or for evaluation. Semantic parsing systems could also be used in lieu of traditional surface form evaluation metrics such as BLEU for machine translation where the evaluation can either be just to check whether the translation is able to produce semantically meaningful sentences or go further and verify whether the result of executing the translated query produces the same denotation.

## Acknowledgements

This survey paper was part of a class project for the Advanced Natural Language Processing course at UMass Amherst offered by Brendan O'Connor. We would like to thank him and Katie Keith for their valuable suggestions and comments.

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
