# OpenReview forum: "A Survey on Semantic Parsing"
_AKBC.ws/2019/Conference — AKBC 2019_

### Official Review · AnonReviewer3 · 2019-01-06
**easy to follow and includes sufficient references**

**Rating:** 7
**Confidence:** 4

**Review:**

The survey paper examines the various components of semantic parsing and discusses previous work. The semantic parsing models are categorized into different types according to the supervision forms and the modeling techniques. Overall, the survey is easy to follow and includes sufficient references. The following points can be improved:

* Training a semantic parser involves NL, MR, context, data, model, and learning algorithms. A summarization and examples of popular datasets are helpful.
* Sec 3.1  Rule based systems can be expanded. The current section is too brief.
* Sec 2.1 Language for Meaning Representation and Sec 2.2 Grammar should be merged.
* P9: "Machine translation techniques" is not a method instead of `` 5. Alternate forms of supervision"
* Sec 8 is too brief now. More discussions on future work are welcome.

minor:
*  Combinatory categorial Grammar ->  Combinatory Categorial Grammar

---

> ### Author Response · Authors · 2019-02-02
> **Addressing suggestions about including datasets and future work**
>
> We thank you for your review and positive comments!
> 1) Yes, we agree that a table summarizing datasets would indeed be helpful. Due to space constraints, we would not have been able to do justice to the large number of datasets available for the task of semantic parsing and consequently planned to have a companion webpage for the survey having a comprehensive table covering datasets. We also plan to include examples from each of the datasets along with links to relevant resources.
>
> 2) Regarding Sec 3.1, we had to omit work due to space constraints especially since we felt it would be more valuable to devote a larger portion of the space to current directions. However, we will try to include some more work in the web page.
>
> 3) "Sec 2.1- Language for Meaning Representation and Sec 2.2 Grammar should be merged."
> We think both the grammar and the language for meaning representations are important and having different subsections highlights their importance. We understand that current neural approaches marginalize out the grammar and can directly generate the language (such as SQL), but there are example of recent works which show significant gain by explicitly using the grammar (e.g. Yin and Neubig, 2017 [1])
>
> 4) * P9: "Machine translation techniques" is not a method instead of `` 5. Alternate forms of supervision"
> We do believe that this setting of using methods inspired by machine translation techniques is unique enough to get its own subsection. The papers described here provide supervision in a manner distinct from the others and we believe that it is worth highlighting.
>
> 5) Re Sec 8: Yes we agree and have added several future directions which we think are important
>
> 6) *  Combinatory categorial Grammar ->  Combinatory Categorial Grammar
> Thanks for pointing that out! We have corrected it in the revision.
>
> References
> [1] A Syntactic Neural Model for General-Purpose Code Generation. Pengcheng Yin and Graham Neubig, ACL 2017

---

### Official Review · AnonReviewer2 · 2019-01-08
**Good survey paper, but no originality**

**Rating:** 7
**Confidence:** 3

**Review:**

Summary:
This paper conducts a thorough survey on semantic parsing, as the title suggested. This paper introduces the formal definition of the semantic parsing, categorized them, describes the development of the system from 1970s to very recent in (fairly) chronological order.

Quality & clarity:
The survey is very thorough and self-contained, and the descriptions are all very clear and well-written.

Originality & significance:
Since it is the survey paper, it’s hard to say it has originality.

General comment:
Although the survey is very thorough, the paper does not have an original contribution, which conference paper should have.


* Update
I had a misunderstanding about the policy regarding survey paper. I agree with other reviewers that the paper is a well-written survey. Therefore I vote for the acceptance.

---

> ### Author Response · Authors · 2019-02-02
> **Motivation for writing this survey and submitting to AKBC**
>
> We thank you for your review and positive comments! The main contributions of this survey are to give a new researcher in the field a comprehensive overview of the various components of a semantic parsing system and to discuss prominent past and current work.
> We believe that the field has evolved so drastically that it is important to have a comprehensive survey that gives an overview of the trends through the years and keeps track of the seminal papers at various stages. We also think that a better understanding of a few open problems will help new researchers work towards concrete research directions. This was the main motivation behind writing this comprehensive survey.
> Also, while we understand that the score was guided by reservations regarding having a survey paper at a conference, we would like to point out that the call for papers mentioned surveys as one of the types of submissions that were being accepted.

---

### Official Review · AnonReviewer1 · 2019-01-10
**Nice review of the development and recent advances in semantic parsing**

**Rating:** 8
**Confidence:** 4

**Review:**

This work provided a comprehensive review of important works in semantic parsing. It starts with the rule-based systems in the early days. Then it described the introduction of statistical methods to learn from natural language and logical form pairs. Finally, it summarized the recent advances in weakly supervised semantic parsing or learning semantic parsing from denotations and the rise of seq2seq models. It also briefly compared different learning strategies (MML, RL, Max-Margin).

The paper is well written and easy to follow. The survey covers most of the important works in the field. It provides a good summary of the development of the field as well as the most recent advances. I support the acceptance of this paper.

Some minor comments:

"the ATIS domain is : What states border Texas :λx.state(x)borders(x, texas)."
Is this example from ATIS? It seems more like a GeoQuery example.

Regarding Reinforcement Learning (section 7.2), there is a recent work (Liang et al, 2018) that is quite relevant. It introduced a principled RL method for semantic parsing, and compared it with other objectives like MML. It also introduced a systematic exploration strategy to address the exploration problem mentioned in this section. Might be worth discussing here.

Memory Augmented Policy Optimization for Program Synthesis and Semantic Parsing, Liang, Chen and Norouzi, Mohammad and Berant, Jonathan and Le, Quoc V and Lao, Ni, Advances in Neural Information Processing Systems, 2018

---

> ### Author Response · Authors · 2019-02-02
> **Addressed minor change + addition of relevant papers**
>
> We thank you for your positive comments. We have updated the paper with your suggestions.
>
> 1) "the ATIS domain is : What states border Texas :λx.state(x)borders(x, texas)."  Is this example from ATIS? It seems more like a GeoQuery example.
> --> It is indeed an example from the GeoQuery dataset. We have updated it accordingly in the revision. Thanks for pointing it out!
> 2) Regarding Reinforcement Learning (section 7.2), there is a recent work (Liang et al, 2018) that is quite relevant.
> --> We absolutely agree with you that Liang et al. 2018 is relevant and we have included it in the survey. Thanks for the suggestion.

---

### Meta-Review · Area_Chair1 · 2019-02-05

**Recommendation:** Accept (Poster)
**Confidence:** 5

**Metareview:**

This is a very nice survey of the history and current state of semantic parsing.  It does a good job covering a very broad field, hitting the right key points along the way.  If there were one thing I would recommend improving, it would be to try to categorize what the open questions are.  There is a short section on "future work", though it simply provides more references to very recent work.  It would be nice to see more detailed thoughts from the authors thoughts on what is missing and what is next, after having read through this vast literature.

Note to PCs: I don't know what to recommend as far as oral vs. poster.  Neither one seems to fit a survey paper very well.  I'm saying poster assuming that only the top few percent of papers will be oral presentations, and while I think this is a good survey, I don't think it is in the top few percent of papers.

---

### Decision · Program_Chairs · 2019-02-15
**AKBC 2019 Conference Decision**

Accept